# Tuning charge transport dynamics via clustering of doping in organic semiconductor thin films

Connor J. Boyle[1,5], Meenakshi Upadhyaya [2,5], Peijian Wang[3], Lawrence A. Renna [1], Michael Lu-Díaz [1], Seung Pyo Jeong[1], Nicholas Hight-Huf [1], Ljiljana Korugic-Karasz[4], Michael D. Barnes[1,3], Zlatan Aksamija [2] & D. Venkataraman [1]

A significant challenge in the rational design of organic thermoelectric materials is to realize simultaneously high electrical conductivity and high induced-voltage in response to a thermal gradient, which is represented by the Seebeck coefficient. Conventional wisdom posits that the polymer alone dictates thermoelectric efficiency. Herein, we show that doping — in particular, clustering of dopants within conjugated polymer films — has a profound and predictable influence on their thermoelectric properties. We correlate Seebeck coefficient and electrical conductivity of iodine-doped poly(3-hexylthiophene) and poly[2,5-bis(2-octyldo-decyl)pyrrolo[3,4-c]pyrrole-1,4(2H,5H)-dione-3,6-diyl)-alt-(2,2′;5′,2″;5″,2‴-quaterthio-phen-5,5‴-diyl)] films with Kelvin probe force microscopy to highlight the role of the spatial distribution of dopants in determining overall charge transport. We fit the experimental data to a phonon-assisted hopping model and found that the distribution of dopants alters the distribution of the density of states and the Kang–Snyder transport parameter. These results highlight the importance of controlling dopant distribution within conjugated polymer films for thermoelectric and other electronic applications.

[1] Department of Chemistry, University of Massachusetts Amherst, 690 N. Pleasant Street, Amherst, MA 01003, USA. [2] Department of Electrical and Computer Engineering, University of Massachusetts Amherst, 100 Natural Resources Road, Amherst, MA 01003, USA. [3] Department of Physics, University of Massachusetts Amherst, 710N. Pleasant Street, Amherst, MA 01003, USA. [4] Department of Polymer Science and Engineering, University of Massachusetts Amherst, 120 Governors Drive, Amherst, MA 01003, USA. [5] These authors contributed equally: Connor J. Boyle, Meenakshi Upadhyaya. Correspondence and requests for materials should be addressed to Z.A. (email: zlatana@umass.edu) or to D.V. (email: dv@chem.umass.edu)

Heat dissipation is ubiquitous—most of the energy we use begins and ends as heat, with more than 60 quads being lost to the environment annually in the US alone[1]. Thermoelectrics (TEs) are solid-state devices that offer reliable and environmentally friendly scavenging of waste heat into electricity. However, their modest efficiency and relatively high cost have hampered their widespread adoption. The conversion efficiencies of inorganic TEs based on crystalline semiconductors have been steadily improving thanks to our growing knowledge of material selection and nanostructuring, aided by first-principles numerical modeling. Organic thermoelectrics (OTEs), on the other hand, are lightweight, flexible, low-cost, and solution-processable, but progress in raising their conversion efficiency has been sporadic and severely hampered by the complexity of their thin film structure and a lack of systematic guidelines for materials discovery and improvement.

TE devices consist of two types of conducting materials, one with holes, which are positive charge carriers, and the other with electrons, which are negative carriers. When one end of a TE is heated, the charge carriers transport heat and move away from the hot junction to the colder end creating a voltage difference. In reverse, when a voltage is applied, the charge carriers transport heat from the cold end to the hot end. Thus, the transport of charge carriers is central to the function of thermoelectric devices. TE energy conversion efficiency is closely related to its dimensionless figure-of-merit, $ZT = \alpha^2 \sigma / \kappa$. Much of the improvement in $ZT$ of inorganic TEs has come from reducing their thermal conductivity, $\kappa$, by alloying and nanostructuring[2–5]. In OTEs, $\kappa$ is inherently low due to lack of long-range periodicity in structure. Therefore further improvements in OTEs must come from the simultaneous increase of the Seebeck coefficient, $\alpha$, which captures the voltage produced per Kelvin of temperature gradient, and electrical conductivity, $\sigma$. A straightforward method to increase electrical conductivities in these materials is to increase the number of charge carriers via chemical doping through oxidation or reduction. Unfortunately, $\alpha$ and $\sigma$ are strongly interdependent. Doping can negatively impact the Seebeck coefficient, generally resulting in a tradeoff between $\sigma$ and $\alpha$. This implies that precise control of the amount of doping is critical for obtaining the maximum power factor, $\alpha^2 \sigma$, and further progress will rely on altering the shape of the $\alpha$ vs. $\sigma$ curve.

Studies have developed empirical relationships between $\alpha$ and $\sigma$ that scale as $\alpha \propto \ln \sigma$ and $\alpha \propto \sigma^{-1/4}$ [6–8], but their physical significance and origin are unclear and thus limiting their utility in the design of OTEs. Recently, Kang and Snyder proposed a generalized two-parameter charge transport model for conducting polymers[9]. They fit the variation in Seebeck coefficient as a function of electrical conductivity to their model to obtain a value for a transport parameter, $s$, and found most polymers to follow a shallow $\alpha$ vs. $\sigma$ trend. The exception was PEDOT:Tos, which exhibited a sharper curvature, leading Kang and Snyder to conclude that the shape of the $\alpha$ vs. $\sigma$ curve is unique to the polymer.

In charge transport, conductivity, $\sigma$, is dictated by the expression: $\sigma = nq\mu$, where $n$ is concentration of charge carriers, $q$ is the carrier charge, and $\mu$ is the charge carrier mobility[10–16]. Determining the charge carrier mobility and its dependence on temperature, field, or carrier concentration in conjugated polymers has led to considerable progress in understanding their charge transport. However, this methodology becomes complicated by the difficulty in determining the carrier concentration with certainty and discrepancies in the reported mobilities for a given material[17,18]. The discrepancies in the reported mobilities arise from differences in device architecture and the measurement technique. Furthermore, measured mobilities are the direct result of a carriers' drift velocity, $\mathbf{v_d}$, at an applied field, $\mathbf{F}$, given by the realtion: $\mathbf{v_d} = \mu \mathbf{F}$. Thus, mobility measurements cannot be used to determine the energy carried per unit carrier or information about the density of states (DOS) that are model and calculation independent.

An effective method to overcome these challenges is to measure both the conductivity and the Seebeck coefficient across a broad range of carrier concentrations. The Seebeck coefficient is the voltage response to an applied temperature gradient, given by the equation: $\alpha = \frac{\Delta V}{\Delta T}$. This methodology takes advantage of the fact that $\alpha$ can be expressed, according to Fritzsche's general equation[19], as $\alpha = \left(\frac{k_B}{q}\right) \int \left(\frac{E - E_F}{k_B T}\right) \frac{\sigma(E)}{\sigma} dE$ and $\sigma$ as $\sigma = \int \sigma(E) dE = q \int g(E) \mu(E) f(E) [1 - f(E)] dE$. In these equations, $E_F$ is the Fermi level and a relative indicator of the level of doping, $g(E)$ is the DOS, and $f(E)$ is the Fermi distribution. This approach is more widely applicable than the simpler Mott formula[20] $\alpha = -\left(\frac{\pi^2}{3}\right)\left(\frac{k_B^2 T}{q}\right) \frac{\partial}{\partial E} \ln[\sigma(E)]|_{E=E_F}$ and has the advantage of connecting $\alpha$ to the average entropy per carrier. Since these expressions are valid (neglecting correlation effects) across all doping levels regardless of the conduction mechanism or the semiconductor's crystalline, semi-crystalline, or amorphous nature, they support numerous mechanisms of conduction in disordered semiconductors[21–23], including hopping models based on the Miller–Abrahams[24] and Marcus[25] jump rates, that add to our physical explanation of charge transport in conjugated polymers and provide structural design criteria for improving their performance.

Generally, both $\alpha$ and $\sigma$ depend on the carrier concentration $n = \int g(E) f(E) dE$ via the Fermi level but they have opposite trends—increasing $n$ fills more states and boosts $\sigma$ but also brings the Fermi level closer to them, decreasing $\alpha$. Thus, there is a narrow range of doping that optimizes the thermoelectric power factor, $\alpha^2 \sigma$, which typically occurs when 10–20% of the states are occupied by a charge carrier. The DOS affects the trade-off between $n$, $\sigma$, and $\alpha$—a sharp DOS separates the $E_F$ from the densely-spaced states, increasing the average energy per carrier $(E - E_F)$ and with it $\alpha$, while a broad DOS implies that the Fermi level moves further to fill the scattered states, flattening the $\alpha$ vs. $\sigma$ curve. Thus, the shape of the DOS has an enormous impact on the scale and the trend of $\alpha$, which is not yet fully understood. Snyder and Kang's recent charge transport model also employs these expressions, substituting for the term $g(E)\mu(E)$ the empirical fitting transport function $\sigma_E(E, T) = \sigma_{E_0}(T) \left(\frac{E - E_t}{k_B T}\right)^s$ which successfully fits a diverse array of conjugated polymers and small molecules using a transport coefficient $\sigma_{E_0}$, comparable to the mobility of the semiconductor, and the transport parameter $s$[9]. Snyder and Kang state that the differences in the $s$ parameter "could be understood as a different 'type' of charge transport" and "may result from the percolation of charge carriers from conducting ordered regions through poorly ordered regions". They speculated that a "search for cases with $s = 1$ could lead to discovery of high-$\sigma_{E_0}$ polymers". Characterizing both $\alpha$ and $\sigma$ across a broad range of doping levels enables a more complete explanation of charge transport that is not possible from measurements of $\sigma$ or mobility alone. Thus far, however, the factors that determine the transport parameter, $s$, in the Synder-Kang model are unclear.

Here, we show that the value of the transport parameter is neither unique to a polymer nor is it confined to specific values; it can be tuned by altering the distribution of the dopants. We show that the way the semiconductor has been doped is fundamentally important to its thermoelectric performance across all carrier concentrations. We also show that the spatial distribution of dopants in the conjugated polymer has a profound impact on the shape of the $\alpha$ vs. $\sigma$ curve and that clustering of dopants in the polymer modifies the shape of the DOS and alters the trend of $\alpha$

vs. $\sigma$ curve. We conclude that the heavy-tailed DOS results in a qualitative change of the $\alpha$ vs. $\sigma$ curve. Therefore, for any given polymer, precise control of amount of doping and the distribution of the dopants are critical for obtaining the maximum power factor. Our work thus opens a straightforward pathway to alter the $\alpha$ vs. $\sigma$ curve, and design efficient organic thermoelectric devices or other organic-based electronic devices.

## Results

**Doping P3HT and PDPP4T with iodine vapor**. Measuring the $\sigma$ and $\alpha$ over a wide range of values requires a large range of doping densities for a particular semiconductor. Chemical[7,8,26–32] or electrochemical oxidation[33] have been used to prepare organic semiconductors at many different doping densities for such characterization, but this requires the preparation of many different samples and the assumption that each sample has a nearly identical morphology. Modulation doping using field-effect transistors[34,35] has been used to measure the $\sigma$ and $\alpha$ at many different, finely controlled doping densities, but the dipole distribution within the dielectric layer can influence the DOS by shifting shallow energy states to a deeper level[22,36], and thus impacts the $\alpha$ vs. $\sigma$ trend in a way that traditional, chemically doped thermoelectric devices would not be affected. Recently, modulation doping using electrochemical field-effect transistors has also been used to control the doping density of thermoelectric materials[37,38]. The manner in which ion infiltration into the conjugated polymer during and after gating the electrochemical field-effect transistor impacts the shape of the DOS currently remains unclear. The methodology we report herein uses spontaneous de-doping of chemically doped conjugated polymers as a rapid and convenient way to capture $\alpha$ vs. $\sigma$ across four orders of magnitude of $\sigma$, using an individual sample, and without gating across a dielectric or electrolytic layer.

Doping organic semiconductors with iodine vapor[39–45] is a well-established strategy to increase the p-type carrier (hole) concentration, resulting in an increased $\sigma$ and decreased $\alpha$. Samples doped in this manner can spontaneously de-dope over time, resulting in a gradual decrease in $\sigma$ and increase in $\alpha$ from their values in the initial doped state. We exploited this de-doping process and measured $\alpha$ as a function of $\sigma$ over a five-orders of magnitude $\sigma$ window. Our method thus captures the trend of $\sigma$ and $\alpha$ using a single sample and without modulation doping.

We used this method to measure the $\alpha$ vs. $\sigma$ relationship over a wide range of $\sigma$ in poly(3-hexylthiophene) (P3HT) and poly[2,5-bis(2-octyldodecyl)pyrrolo[3,4-c]pyrrole-1,4(2H,5H)-dione-3,6-diyl]-alt-(2,2′;5′,2′′;5′′,2′′′-quaterthiophen-5,5′′′-diyl)] (PDPP4T), two widely-studied semiconducting conjugated polymers. By exposing the films to iodine vapor, the polymer films acquire a strongly spatially heterogeneous and reversible chemical oxidation (de-doping) (Fig. 1) that changes both conductivity and Seebeck coefficient over a time scale of ~4–24 h.

Doping by iodine vapor was performed by exposing the polymer films to iodine vapor (Methods section) for 2 h at either 25 °C or 75 °C. We then measured, in situ, $\alpha$ and $\sigma$ as the films underwent spontaneous de-doping. In each sample, $\alpha$ increased and $\sigma$ decreased over time as the film de-doped (Fig. 2), as is expected for organic semiconductors with decreasing carrier concentration. Both $\alpha$ and $\sigma$ depend on the free carrier concentration, but this concentration is difficult to determine because of the spontaneous de-doping process and the absence of electron paramagnetic resonance signal from bipolaron carriers. Plotting the Seebeck coefficient vs. the conductivity measured at each time interval on the log-log scale illustrates the comparison of the charge transport properties of these samples while avoiding the need to measure carrier concentration (Fig. 3). The shape of

the $\alpha$ vs. $\sigma$ curve for P3HT is similar for the sample doped at 25 °C and at 75 °C, and both curves appear to be consistently flat with a low Seebeck coefficient. In contrast, the shape of the $\alpha$ vs. $\sigma$ curve for PDPP4T changed with the doping temperature. Next, to isolate the effect of doping temperature from possible annealing effects, we sequentially annealed PDPP4T at 75 °C and then doped the sample at 25 °C. The $\alpha$ vs. $\sigma$ trend for the annealed sample was similar to the un-annealed polymer that was doped at 25 °C (Supplementary Fig. 1), indicating that thermal annealing at 75 °C alone may not be cause of the modified $\alpha$ vs. $\sigma$ trend. We additionally found that repeatedly doping the same sample of PDPP4T at 25 °C, de-doping it, and re-doping it recovered the same $\alpha$ vs. $\sigma$ trend (Supplementary Fig. 2). To verify the unusual dependence of doping temperature on the thermoelectric properties of PDPP4T, we repeated the doping and measurement at temperatures of 25 °C and 75 °C using PDPP4T synthesized in-house, and observed similar behavior (Supplementary Fig. 3).

In Kang and Snyder's model, a flatter and more gradual curve is indicative of a transport parameter of $s = 3$, while a curve maintaining a greater $\alpha$ until a sudden, sharp drop-off at high $\sigma$ is consistent with $s = 1$. We applied the Kang and Snyder model to our experimental data and surprisingly did not achieve the best fit using either $s = 1$ or $s = 3$ (Supplementary Fig. 4). We instead focused our attention to the effect of doping on the $\alpha$ vs. $\sigma$ trend, since doping results in coulombic potentials distributed throughout the polymer and these potentials alter the DOS and therefore the $\alpha$ vs. $\sigma$ trend. Most studies assume that doped semiconductor films are homogenous[46]. We reasoned there is an alternate possibility: the dopants in the films may be heterogeneously distributed in the thin films as nanoscopic clusters similar to the nanoscopic charged regions in a contact electrified sample[47]. Dopant clusters have been observed in inorganic semiconductors[48] but are not commonly invoked in organic semiconductors as they are often doped in solution[49]. We then hypothesized that doping temperature alters the clustering of the dopants and thus the shape of the DOS, which in turn alters the relationship between the Seebeck coefficient and electrical conductivity. To test this hypothesis, we probed the clustering and spatial heterogeneity of dopants in the conjugated polymer films using KPFM. We characterized the polymer films using photoluminescence spectroscopy (Supplementary Fig. 5) for doping and X-ray scattering to identify any structural changes upon doping. We also computed the effect of the dopant-induced distribution of the DOS on the $\alpha$ vs. $\sigma$ curve (Supplementary Fig. 6).

**X-ray scattering studies**. The wide angle X-ray Scattering (WAXS) patterns for pristine and unannealed P3HT showed two signature peaks at q values of 0.37 Å$^{-1}$ ($d_{100} = 16.98$ Å), and 1.65 Å$^{-1}$ ($d_{020} = 3.81$ Å) (Supplementary Fig. 7) and is consistent with literature values. Annealing the pristine films at 75 °C results in slight peak shifts with peaks appearing at q = 0.38 Å$^{-1}$ ($d_{100} = 16.53$ Å) and 1.67 Å$^{-1}$ ($d_{020} = 3.76$ Å). For unannealed pristine PDPP4T, we observed two signature peaks at $q$-values of 0.30 Å$^{-1}$ ($d_{100} = 20.94$ Å), and 1.66 Å$^{-1}$ ($d_{020} = 3.79$ Å). For films annealed at 75 °C, the peaks appear at q values of 0.30 Å$^{-1}$ ($d_{100} = 20.94$ Å), and 1.67 Å$^{-1}$ ($d_{020} = 3.76$ Å). Both polymers had broad peak around q = 1.25 Å$^{-1}$ attributed to the amorphous phase. We did not observe additional peaks in thermally annealed samples.

The WAXS pattern of a P3HT film doped at 25 °C was identical to the pattern of a pristine P3HT film (Supplementary Fig. 8a) indicating that the dopants may reside in the amorphous regions[50,51]. The WAXS pattern of the de-doped film was also identical to the pattern of a pristine film. The WAXS pattern of P3HT films doped at 75 °C shows peaks at q = 0.35 Å$^{-1}$ ($d_{100} = 17.95$ Å) and at 1.73 Å$^{-1}$ ($d_{020} = 3.63$ Å) indicating that the

**Fig. 1** Schematic of doping and de-doping of conjugated polymers using iodine. **a** poly(3-hexylthiophene) P3HT and **b** poly[2,5-bis(2-octyldodecyl)pyrrolo [3,4-c]pyrrole-1,4(2H,5H)-dione-3,6-diyl)-alt-(2,2';5',2'';5'',2'''-quaterthiophen-5,5'''-diyl)] (PDPP4T) are chemically doped in the presence of iodine vapor, but are unstable and gradually de-dope in the absence of iodine vapor

dopants may have penetrated the crystalline domains (Supplementary Fig. 8b). The broad peak around $q = 1.25\,\text{Å}^{-1}$ also narrowed and has a pronounced feature. The WAXS patterns of PDPP4T films doped at room temperature and de-doped films were identical to pristine PDPP4T (Supplementary Fig. 8c). The WAXS pattern of PDPP4T films doped at 75 °C shows peaks at $q = 0.28\,\text{Å}^{-1}$ ($d_{100} = 22.44\,\text{Å}$) and at $1.68\,\text{Å}^{-1}$ ($d_{020} = 3.74\,\text{Å}$) (Supplementary Fig. 8d). Overall, the shifts were smaller in the WAXS patterns of PDPP4T films compared to P3HT films doped at different temperatures. The WAXS patterns also indicate that the dopant distribution in the films doped at 25 °C are different from films doped at 75 °C. To understand the differences between the doped films, we turned to KPFM.

**Kelvin probe force microscopy and photoluminescence studies**. KPFM exploits the capacitive interaction between a metal (Pt/Ir)-coated probe and the sample. This interaction is associated with the work function difference between the probe and the sample, manifested as the surface potential contrast or SPC[52]. Since chemical doping of the conjugated polymer by iodine vapor alters the polymer's work function and thus the SPC, KPFM can track and map the changes that occur upon doping along the film's surface. We mapped the SPC of films of P3HT and PDPP4T doped at 25 °C and 75 °C. In the SPC maps, we observed the presence of dense regions in the polymer films that had a different SPC than the surrounding sections. Photoluminescence microscopy imaging (Supplementary Fig. 5) of a similar P3HT film doped at 25 °C displays dense regions of dark P3HT, consistent with chemically oxidized P3HT[53], with similar cross-sectional areas surrounded by brighter-emitting P3HT. Therefore, we inferred that the dense regions in the SPC map correspond to densely doped states on the film's surface surrounded by less densely doped P3HT.

We then determined the homogeneity of the distribution of iodine-doped states within the film using the width of the SPC distributions. P3HT films doped at 25 °C displayed a remarkably wide distribution of SPC and regions of densely doped states with cross-sectional areas on the order of $1\,\mu\text{m}^2$, indicating the distribution of iodine-doped states is heterogeneous, while pristine P3HT has a narrow distribution of SPCs (Fig. 4). The

appearance of an exponential tail to this SPC distribution is consistent with the exponential DOS distribution we simulated to fit the $\alpha$ vs. $\sigma$ curve for P3HT doped at 25 °C. P3HT films doped at 75 °C showed a marginally narrower distribution of SPCs, suggesting that P3HT films doped at 75 °C exhibit a slightly more homogeneous distribution of iodine-doped states compared to P3HT doped at 25 °C. PDPP4T doped at 25 °C displayed a significantly narrower SPC distribution than either P3HT sample, and PDPP4T doped at 75 °C had yet an even narrower SPC distribution. The trend of SPC distributions for each sample is consistent with the DOS distributions we modeled, confirming that the $\alpha$ vs. $\sigma$ curve can be fit to an appropriate DOS by solving the Pauli master equation. PDPP4T is doped with a more homogeneous distribution of iodine-doped states than P3HT and homogeneity of iodine-doped states increases with increasing doping temperature.

We tracked the SPC distribution of P3HT doped at 25 °C over time as the sample spontaneously de-doped to determine the effect of de-doping on the dopant distribution homogeneity (Fig. 5a). The SPC distributions recorded at each time were fit to a Gaussian distribution (Fig. 5b) so that the dopant homogeneity can be described in terms of the mean (Fig. 5c) and width of this distribution (Fig. 5d). The width of the SPC distribution decreases over time, indicating the dopant distribution becomes more homogeneous as the sample de-dopes and the concentration of dopant counterions in the film decreases. The mean of the Gaussian fit of the SPC distribution becomes more positive over time, consistent with the increase in $E_F$ of P3HT upon de-doping.

**Modified Gaussian phonon-assisted hopping model**. We calculated $\alpha$ and $\sigma$ by numerically solving the Pauli master equation (PME) that describes phonon-assisted carrier hopping between localized sites (see Methods for simulation details) whose energies were sampled from the carrier DOS. We obtained $\alpha$ and $\sigma$ at various carrier densities by varying the Fermi level $E_F$ further and closer to the center of the energy distribution, analogous to doping. Previous studies have used a Gaussian distribution to describe the DOS, where the width of the DOS accounts for the degree of energetic disorder in the structure[54–57]. However, we find that in such a model, varying the energetic disorder simply

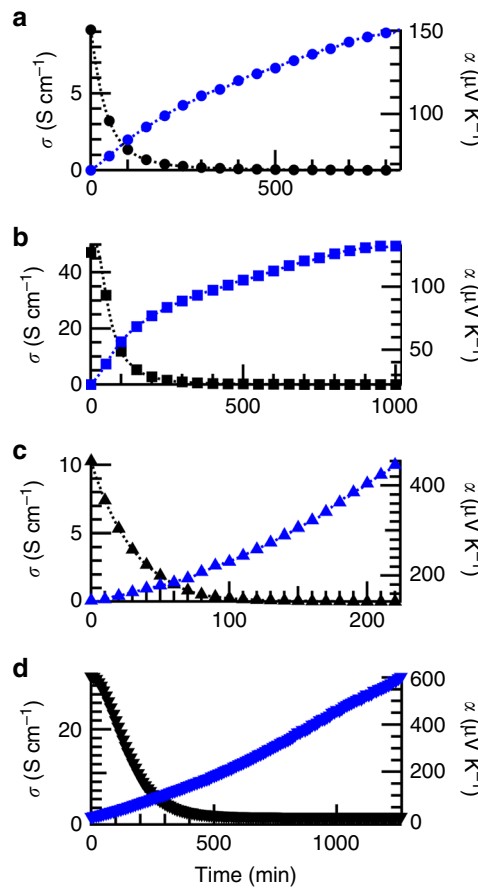

**Fig. 2** Conductivity and Seebeck coefficient of polymers measured during de-doping. Conductivity (black markers, left axes) and Seebeck coefficient (blue markers, right axes) for **a** poly(3-hexylthiophene) (P3HT) doped at 25 °C (circles), **b** P3HT doped at 75 °C (squares), **c** poly[2,5-bis(2-octyldodecyl)pyrrolo[3,4-c]pyrrole-1,4(2H,5H)-dione-3,6-diyl)-alt-(2,2′;5′,2′′;5′′,2′′′-quaterthiophen-5,5′′′-diyl)] (PDPP4T) doped at 25 °C (up triangles), and **d** PDPP4T doped at 75 °C (down triangles). One of every five data points collected is plotted along a dashed line as a visual guide

shifts the α vs. σ on the log-log curve with minimal difference in its shape (Fig. 6b) and cannot fully account for the significant difference in the slope of the α vs. σ plots between the P3HT and PDPP4T plots. It has been shown that long-range coulombic interaction between the ionized dopant molecules and the localized carriers further increases energetic disorder and broadens the deep tail of the DOS[58]. The physical distribution of dopant molecules within the sample and the size of the dopant clusters both further intensate the impact on the DOS. We calculate the DOS resulting from doping concentration $N_d$ in clusters having size $C_s$, according to Eq. 2 in the Methods section, and find that doping and clustering result in a heavy-tailed distribution with a Gaussian core and a wide quasi-exponential tail (Fig. 6a). To particularly examine the effect of the exponential tail, we compare the α vs. σ plot for a Gaussian and a purely exponential DOS in Fig. 6b. The exponential DOS results in a much flatter α vs. σ curve, which can be understood from the Mott formula[59]

$$\alpha \propto \frac{d\ln[g(E)]}{dE} + g(E)\frac{d[\mu(E)]}{dn} \qquad (1)$$

When the $\mu(E)$ is only weakly dependent on carrier concentration, the second term in Eq. (1) is small and an exponential DOS $g(E) \propto \exp(-E/\Gamma_E)$ leads to a nearly constant α independent of doping or σ but inversely proportional to the energetic disorder $\Gamma_E$ that dictates the width of the DOS.

More realistically, Fig. 6c compares the effect of a Gaussian DOS distribution vs. a heavy-tailed DOS distribution computed with several values of $N_d$ and $C_s$. We compare our simulated results to Snyder and Kang's charge transport model[9] and find that a Gaussian distribution results in significantly lower transport parameter $s \leq 1.5$ compared to the exponential case which has $s \geq 1.5$, and this difference increases further with increasing energetic disorder. However, at high energetic disorder in the presence of a heavy tail the curve cannot be fit by transport parameter values of $s = 1$ or $s = 3$ (Fig. 6c and Supplementary Fig. 6), indicating the limitations of a band model in predicting transport in highly disordered systems (Supplementary Note 2). Comparing our hopping model simulations to experimental data from Fig. 3, we find that PDPP4T doped at 75 °C is most closely fit with α vs. σ computed from a purely Gaussian distribution

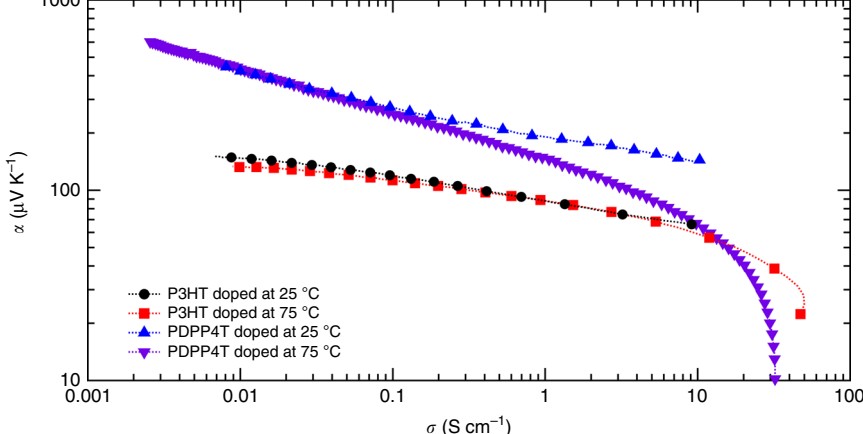

**Fig. 3** Seebeck coefficient vs. conductivity plot. A log-log plot of Seebeck coefficient vs. conductivity for poly(3-hexylthiophene) (P3HT) doped at 25 °C (black circles), P3HT doped at 75 °C (red squares), poly[2,5-bis(2-octyldodecyl)pyrrolo[3,4-c]pyrrole-1,4(2H,5H)-dione-3,6-diyl)-alt-(2,2′;5′,2′′;5′′,2′′′-quaterthiophen-5,5′′′-diyl)] (PDPP4T) doped at 25 °C (blue up triangles), and PDPP4T doped at 75 °C (purple down triangles). One of every five data points collected is plotted along with a dashed line as a visual guide

   **5**

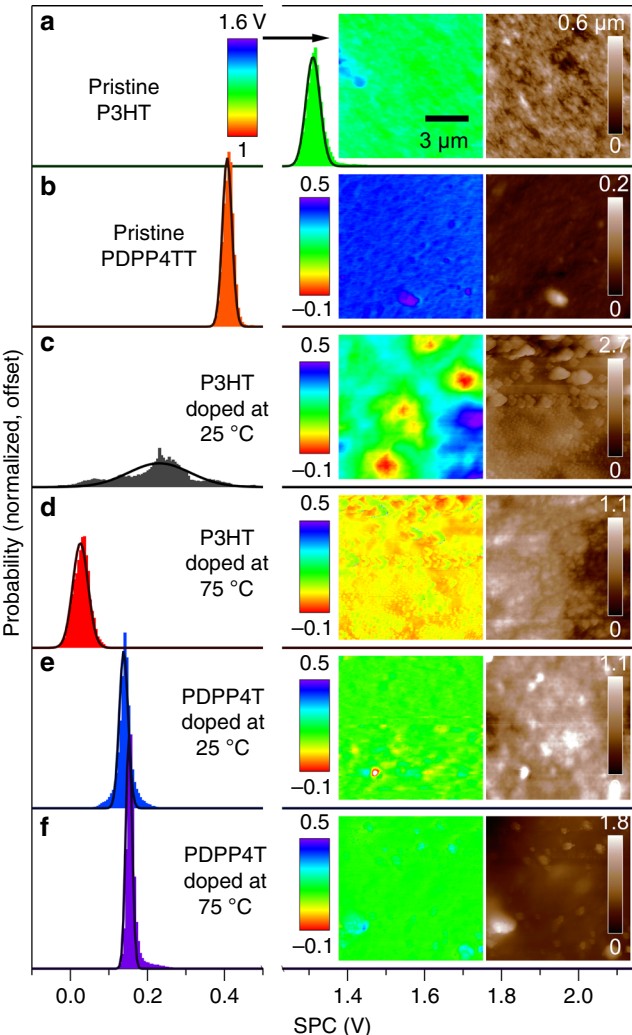

**Fig. 4** Impact of doping on surface potential contrast. A histogram of measured surface potential contrast (SPC) and its Gaussian fit, with the height map and SPC map inset, of **a** pristine poly(3-hexylthiophene) (P3HT), **b** pristine poly[2,5-bis(2-octyldodecyl)pyrrolo[3,4-c]pyrrole-1,4 (2H,5H)-dione-3,6-diyl]-alt-(2,2′;5′,2′′;5′′,2′′′-quaterthiophen-5,5′′′-diyl)] (PDPP4T), **c** P3HT doped at 25 °C, **d** P3HT doped at 75 °C, **e** PDPP4T doped at 25 °C, and **f** PDPP4T doped at 75 °C

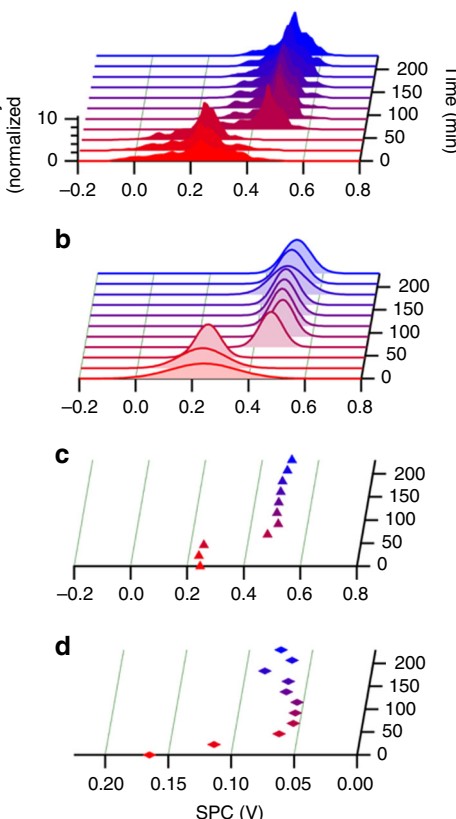

**Fig. 5** Changes in surface potential contrast as the polymers de-dope. **a** Histograms of the surface potential contrast distribution, **b** Gaussian fit to each histogram, and the **c** mean and **d** width of the Gaussian fits of a film of poly(3-hexylthiophene) (P3HT) initially doped at 25 °C as it spontaneously de-dopes

with a width of 10 $k_B T$, whereas PDPP4T doped at 25 °C and P3HT is best fit with a dopant-induced heavy-tailed DOS that gives us the required change in slope (Fig. 6d).

To further elucidate the relationship of the shape of the $\alpha$ vs. $\sigma$ curve to the shape of the DOS, we compare schematically the Gaussian and the heavy-tailed DOS in Fig. 7a and b. There we show states filled up to the Fermi level using different colors, while Fig. 7c shows the Seebeck coefficients $\alpha$ corresponding to those colors. For a given $\alpha$ vs. $\sigma$ curve, as $E_F$ approaches the center of the DOS and states are filled the average energy per carrier $(E - E_F)$ decreases, and with it $\alpha$. For the purposes of qualitative analysis of trends, the Seebeck coefficient can be approximately related to the slope of the DOS via the Mott formula in Eq. 1. Focusing on the first term which is typically dominant, a larger slope in the logarithm of the DOS $g(E)$ implies a larger Seebeck coefficient; this can be observed in the red region "1" in Fig. 7a, b. As the Fermi level approaches the middle of the DOS (region "4"), represented by $E = 0$ in our calculations, the slope approaches zero as does the Seebeck coefficient (Fig. 7c). We find that this part of the curve is always fit by $s \leq 0.5$

(Supplementary Fig. 6) as the center of the DOS is symmetric inside the Fermi window $\left(-\frac{\partial f}{\partial E}\right)$ irrespective of the presence of a heavy tail. Thus we conclude that it is this 'change in shape' of the DOS that instigates the change in slope in the $\alpha$ vs. $\sigma$ plot.

Although, the presence of a heavy tail leads to a constant $\alpha$ and a flatter $\alpha$ vs. $\sigma$ curve advantageous for thermoelectrics, the Coulomb interaction causing the heavy tail also increases the energetic disorder, which has an adverse effect on $\alpha$ (see comparison of a 3 $k_B T$ Gaussian and heavy-tailed DOS in Fig. 7c). Hence, a narrower DOS and a smaller transport parameter $s$ is more advantageous for thermoelectric applications. This impact of an energetically disordered, heavy-tailed DOS on the thermoelectric properties is consistent with that recently reported by Kemerink and co-workers[46]. While Kemerink and co-workers assumed a homogenous distribution of the dopants and use this to capture the experimentally determined $\alpha$ vs. $\sigma$ curve, we find that the spatial heterogeneity of dopants is necessary to describe the impact of doping on the shape of the DOS.

## Discussion
Our studies establish that the shape of the $\alpha$ vs. $\sigma$ curve depends on the clustering of the dopants in the conjugated polymer. The dopant distribution affects the carrier DOS, with dopant clustering dramatically increasing the energetic disorder, which in turn affects the charge transport properties. We associate flattened $\alpha$ vs. $\sigma$ trends with heterogeneous spatial distributions of dopants throughout the sample using surface potential contrast (SPC) mapping by Kelvin probe force microscopy (KPFM), which

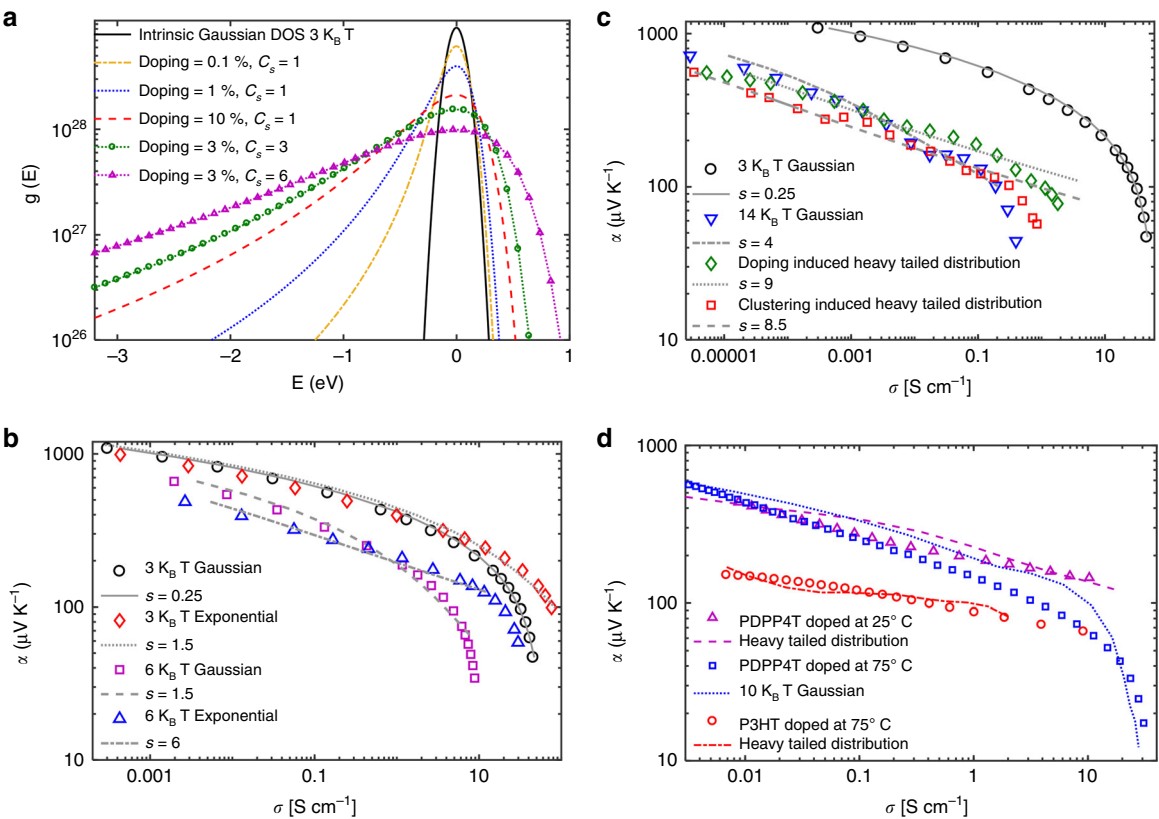

**Fig. 6** Simulated dopant-induced Seebeck and conductivity trends. **a** The effect of doping and clustering on the density of states (DOS) distribution with intrinsic Gaussian width of 3 $k_BT$. The charge carriers interact coulombically with the ionized dopants creating additional deep trap states, resulting in a heavy-tailed DOS. Log-log plot of Seebeck coefficient vs. conductivity showing the change in slope due to: **b** Gaussian and exponential DOS, **c** Doping and clustering induced heavy-tailed DOS. The doping-induced distribution is computed with dopant concentration $N_d = 10\%$ and cluster concentration $C_s = 1$, and the clustering induced distribution with $N_d = 3\%$ and $C_s = 3$. We have fit our simulated results (symbols) to Snyder and Kang's charge transport model (gray lines) and the corresponding transport parameter 's' values are shown. **d** Comparison of our model to experimental data from Fig. 3. ($N_d = 4\%$ and $C_s = 1$ for pink dashed line, $N_d = 0.9\%$ and $C_s = 9$ for red dot-dashed line, $\gamma = 0.01$ and $\Sigma_{ij} = 0.0025$ for all three cases)

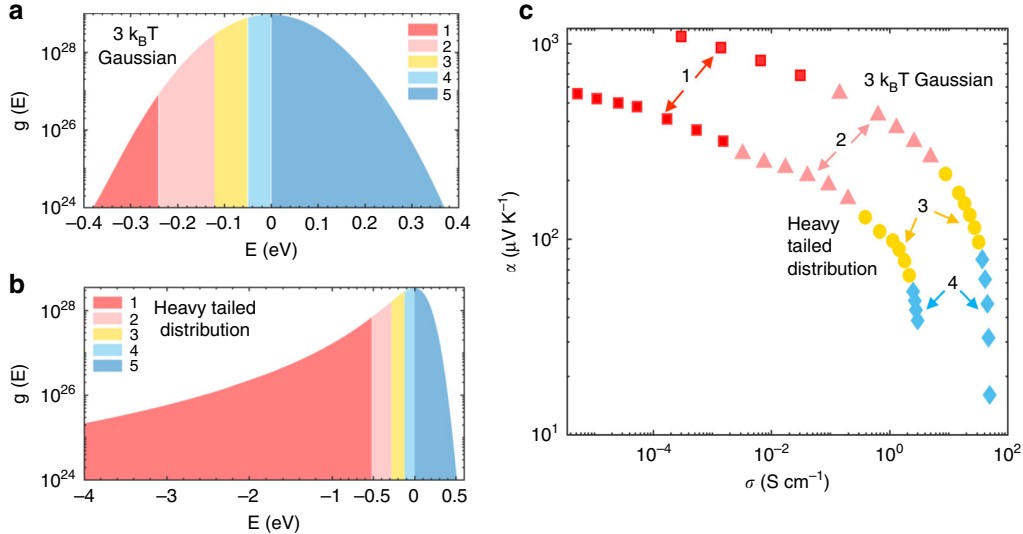

**Fig. 7** Impact of density of states distribution on Seebeck coefficient vs. conductivity curve. Schematic showing the filling up of the **a** Gaussian and **b** heavy-tailed density of states DOS with increasing doping, by varying the Fermi level $E_F$ further and closer to the center of the energy distribution and **c** the corresponding effect on the log-log plot of Seebeck coefficient vs. conductivity

suggests that a heterogeneous distribution of coulombic potentials from doping broadens the DOS and thus depresses the Seebeck coefficient. We then calculated the doping-induced DOS and simulated carrier transport using a modified Gaussian disorder model and the Pauli Master equation to explain how the shape of the DOS impacts the $\sigma$ and $\alpha$ trend. Based on these studies, we show that the way the semiconductor has been doped is fundamentally important to its thermoelectric performance across all carrier concentrations. We show that the spatial distribution of dopants in the conjugated polymer has a profound impact on the shape of the $\alpha$ vs. $\sigma$ curve and that clustering of dopants in the polymer modifies the shape of the DOS and alters the trend of $\alpha$ vs. $\sigma$ curve.

We found that PDPP4T exhibited two very distinct shapes of the $\alpha$ vs. $\sigma$ curve, depending only on the temperature at which doping was performed. KPFM measurements uncovered that room-temperature doping resulted in dopant inhomogeneity and clustering. We simulated the effect of clustering in a phonon-assisted hopping model of charge transport and found that it caused a modification of the electronic density of states (DOS) which resulted in a heavy tail, while homogenous doping maintained a DOS consistent with the Gaussian disorder model[15]. We conclude that the heavy-tailed DOS results in a qualitative change of the $\alpha$ vs. $\sigma$ curve.

Our studies suggest that tuning the energetic disorder by controlling the dopant counterion distribution within the doped film of the organic semiconductor can lead to substantial improvements in the thermoelectric performance of organic semiconductors. The conventional wisdom is that semi-crystalline polymers with crystalline domains may be better for OTEs. Our studies show that a uniform dopant distribution will lead to narrow energetic disorder with enhanced thermoelectric properties. Thus, we predict that organic systems with molecular packing or morphologies that can accommodate uniform doping may be superior candidates for OTEs. Our studies also illustrate the need to understand the role of crystalline and amorphous phases in polymer films, frontier orbital energies, kinetics of dopant diffusion, and dopant-polymer miscibility[7,29,38,60,61] on dopant clustering. Thus, controlling dopant clustering in organic semiconductors will be necessary to improve the existing and designing the next generation of organic electronic applications.

## Methods

**Materials.** P3HT ($M_w = 90$ kDa, 96% HT regioregularity) was purchased from Rieke Metals and PDPP4T ($M_w = 171,138$ Da, $Đ = 2.45$) was purchased from Ossila. Chloroform and iodine were purchased from commercial vendors and used as received.

**Film preparation.** Solutions of 5 mg/mL P3HT or 8 mg/mL PDPP4T in chloroform were stirred for no less than 4 h before drop-casting. 0.23 mL of the polymer solution was drop cast onto a pre-cleaned, handcut, 1.1 cm × 2.2 cm glass coverslip that was preheated to 45 °C on a hot plate, and this was immediately covered with a watch glass to impede the escape of chloroform vapors and slow the rate of evaporation during drop-casting. After 10 min, the sample heating element was turned off, and the sample was let stand under ambient conditions for no less than 24 h to evaporate the chloroform further. P3HT films were ~4 μm thick and PDPP4T films were ~9 μm.

**Doping with iodine vapor.** 50 ± 5 mg iodine crystals were loaded into a 1 mL glass vial, and this vial was loaded into a 20 glass mL vial. For samples doped at 25 °C, the 20 mL vial was first capped and let stand for no less than 12 h to allow solid-vapor equilibration of the iodine within the vials. For samples doped at 75 °C, the 20 mL vial was first capped and let stand in a 75 °C oven for 10 min to allow solid-vapor equilibration of the iodine within the vials while the polymer film was simultaneously heated to 75 °C for 10 min. In each case, the iodine doping was carried out by placing the polymer film into the 20 mL vial (which contained iodine vapor and the 1 mL vial of 50 ± 5 mg iodine), capping the vial, and heating to the specified temperature for 2 h.

**Characterization of thermoelectric properties.** Samples were transferred from their iodine doping chamber to a custom-built (reported elsewhere[45,58] and briefly summarized herein) thermoelectric characterization apparatus in a timely fashion since they began de-doping immediately and rapidly in the absence of iodine vapor. The sample was placed on an insulating glass slide bridging one heated copper block and one unheated copper block to establish a temperature gradient. A PTFE block containing four platinum wire electrodes in a linear arrangement and two k-type thermocouples was clamped onto the sample. The sense probes and thermocouples were separated by a distance of 1.4 cm. This entire apparatus was enclosed within an electrically grounded metal box. A LabView program was used to interface with a digital dual input thermometer for the k-type thermocouples, a Keithley 2182 A nanovoltmeter, and a Keithley 2440 5 A source meter to repeat measurements of the temperature gradient $\Delta T$, voltage gradient $\Delta V$, and $I–V$ characteristics respectively across the sample sequentially and repeatedly every 10 min for P3HT and every 2 min for PDPP4T. A $\Delta T$ of ~20 °C with an average temperature of ~50 °C was applied to P3HT, and a temperature gradient of ~10 °C with an average temperature of ~45 °C was applied to PDPP4T. $\Delta V$ was taken to be the average of 1000 voltage measurements from the Keithley 2182 A nanovoltmeter clamped to the sense probes, and $\alpha$ was taken to be $\Delta V/\Delta T$. Only measurements for which the standard deviation of the 1000 voltage measurements is less than 1% of their mean are used to calculate $\alpha$, while the remaining measurements were discarded, to ensure only reliable estimates of $\alpha$ are considered. The conductance was determined from the slope of $I−V$ curve obtained by the Keithley 2440 5 A using a four-probe measurement.

**Simulation details.** Our charge transport model is based on electron hopping between localized sites and the hopping rate between sites ($i–j$) is calculated from the Miller–Abrahams rate equation[24] $W_{ij} = \nu_0 exp\left[-2\gamma_{ij}R_{ij}\right]\left[N\left(\Delta E_{ij}\right) + \frac{1}{2} \pm \frac{1}{2}\right]$, where $\nu_0 = 5 \times 10^{12}$ s$^{-1}$ is the attempt to escape frequency, $\gamma = 1$ is the overlap factor ($\gamma_{ij} = \gamma_i + \gamma_j$, $\gamma_i$ and $\gamma_j$ are the site-specific contributions obtained from a Gaussian distribution of width $\Sigma_{ij} = 0.25$) and $R_{ij}$ is the distance between the sites. $N(E)$ is the Bose-Einstein distribution with $+\frac{1}{2}$ for hops upwards in energy ($E_i > E_j$) by absorption of a phonon and $-\frac{1}{2}$ for downward hops with the emission of a phonon. $\Delta E_{ij} = E_j - E_i - e\mathbf{F}\Delta R_{ij,\mathbf{x}}$ where, $E_i$ and $E_j$ are the energies of the sites and $\mathbf{F} = 10^6$ Vm$^{-1}$ is the externally applied electric field. These are the values used throughout the simulation unless stated otherwise. We simulate a $35 \times 25 \times 25$ lattice of sites with an average distance between adjacent sites $a = 0.38$ nm, and consider up to the third-nearest neighbor, which implies a maximum hopping distance of $\sqrt{3}a$.

We numerically solve the Pauli master equation to compute the time-averaged occupational probabilities of the sites using a non-linear iterative solver, and the initial site occupation probability is given by the Fermi-Dirac distribution[62]. In steady-state, $\frac{dp_i}{dt} = 0 = \Sigma_j[W_{ij}p_i(1 - p_j) - W_{ji}p_j(1 - p_i)]$ where, $p_i$ is the occupation probability of a site $i$ and $W_{ij}$ is the hopping transition rate, and the whole term is summed over the neighbor sites $j$. The current density $J$ is found by a summation over all the carriers in the direction of the applied field, $J = \frac{e}{a^3 N}\sum_{i,j} W_{ij}p_i(1 - p_j)R_{ij,\mathbf{x}}$ and the Seebeck coefficient is calculated as $\alpha = \frac{E_F - E_t}{eT}$ where, $E_t$ is the average transport energy, calculated from $E_t = E_i = \frac{\sum_{i,j} E_i W_{ij}(1-p_j)R_{ij,\mathbf{x}}}{\sum_{i,j} W_{ij}p_i(1-p_j)R_{ij,\mathbf{x}}}$[54,57,63].

Arkhipov et al.[61], have shown that Coulomb interaction between ionized dopants results in a heavy-tailed DOS given by $g(E) = \frac{4\pi q^6 N_d}{(4\pi\varepsilon_0\varepsilon)^3}\int_{-\infty}^0 \frac{dE_c}{E_c^4} exp\left[\frac{4\pi N_d}{3}\frac{q^6}{(4\pi\varepsilon_0\varepsilon E_c)^3}\right]g_i(E - E_c)$ where, $N_i$ and $N_d$ are the intrinsic and dopant concentration respectively, $E_c$ is the potential energy of the Coulomb interaction and $g_i$ is the intrinsic Gaussian DOS centered at 0 energy and given by $g_{i(E)} = \frac{1}{2\pi\Gamma_E^2}\exp\left(-\frac{E^2}{2\Gamma_E^2}\right)$. However, they do not consider the impact of dopants clustering. In the presence of dopant clustering, the probability density $w(r)$ of the minimum distance at which a dopant cluster is present can be modeled by a Poisson distribution $w(r) = 4\pi r^2 N_s e^{\left(\frac{4\pi}{3}N_s r^3\right)}$ where, $N_s$ is the cluster density. The potential energy of the Coulomb interaction between the localized charge carrier and dopant cluster is now $E_c = -C_s e^2/4\pi\varepsilon_0\varepsilon r$ where, $C_s$ is the number of dopants in each cluster. Combining these equations to obtain the energy distribution of localized states over the intrinsic energy $E_i$ and $E_c$ we get

$$g(E) = \frac{4\pi q^6 N_s C_s^3}{(4\pi\varepsilon_0\varepsilon)^3}\int_{-\infty}^0 \frac{dE_c}{E_c^4} exp\left[\frac{4\pi N_s C_s^3}{3}\frac{q^6}{(4\pi\varepsilon_0\varepsilon E_c)^3}\right]g_i(E - E_c) \qquad (2)$$

where, $N_s = \frac{N_d}{C_s}$. We use the rejection sampling technique to generate an energy distribution from the calculated DOS; and then randomly assign an energy to each site from the distribution. Details about solving the non-linear PME has been described in Supplementary Note 1 and further details of our model has been reported in an earlier work[58].

**X-ray Scattering, KPFM, and PL instrumentation and characterization.** The microstructural characterization of the polymer films was conducted in a SAX-SLAB Ganesha 300XL X-ray scattering instrument equipped with a Xenocs GeniX

3D CuKα source (λ = 0.15418 nm) and a Dectris Pilatus 30 K photon-counting detector. Ultra-high vacuum (<2E − 1 mbar) was applied to reduce background scattering. WAXS measurements were obtained in transmission mode, and the sample-detector distance was ~100 mm.

KPFM experiments were conducted on a Digital Instruments Bioscope Atomic Force Microscopy (AFM), with AppNano ANSCM-PA Pt/Ir-coated Si cantilever probe. The probe possesses a resonance frequency of around 254 kHz. A mixed AC and DC voltage electrical excitation signal was applied between the probe, lifted by 40 nm, and the grounded sample, to acquire the surface potential contrast. The scan rate was set to be 0.4 Hz, and the sampling density was 512 lines and 512 samples per line.

The PL imaging was conducted with a Princeton Instrument Acton Photomax 512 EMCCD camera, which was cooled down thermoelectrically to −70 °C. The images were captured at an exposure time of 0.2 s.

## Data availability

All the data presented in this paper and in the supporting information are available from the authors upon request. The computer codes used in this work are available from the authors upon request. All the data presented in the figures in the manuscript can be downloaded as a ZIP file from URL: https://doi.org/10.7275/7fkz-k161.

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

## Acknowledgements
The authors thank Dr. Stephen Dongming Kang and Prof. G. Jeffrey Snyder for discussions on fitting to their charge transport model and sharing their model-fitting code. The authors thank Profs. Frank E. Karasz and Paul M. Lahti for the thermoelectric measurement equipment. The authors thank Profs. Feng Liu and Thomas P. Russell for providing home-made PDPP4T used to verify the trends found in commercially sourced PDPP4T. Z.A. thanks Mr. Timothy Mirabito for fruitful discussion about the phonon-assisted hopping model.

## Author contributions
D.V. and Z.A. conceived the idea. D.V. and C.J.B. designed and implemented the experimental setup for conductivity and Seebeck coefficient characterization. Z.A. and M.U. implemented the phonon-assisted hopping charge transport simulation code. M.U. performed the phonon-assisted hopping charge transport simulations and DOS calculations. M.D.B., P.W., and N.H.-H. measured the KPFM and PL microscopy. L.A.R. updated the code for fitting to the empirical charge transport model. S.P.J. and M.L.-D. performed X-ray scattering analysis the polymer films. L.K.-K. described how structural disorder in polymers is expected to impact the thermoelectric charge transport.

## Additional information

**Competing interests:** The authors declare no competing interests.

