## [Peer Review File · Nature Communications]

Reviewers' comments:

Reviewer #1 (Remarks to the Author):

This study described a story about the effects of aggregation of dopant on charge transport in conducting polymers. The morphology of doped films could be tuned by doping temperature. The aggregation of dopant broadened the distribution of DoS, and hence affecting charge transport. The spatial distribution of dopant was highlighted to be an important role in thermoelectrics and other electronic applications.

The novelty of this work is to probe the spatial heterogeneity and the energy distribution by using KPFM, which provides an idea on characterization for future study. Additionally, the design of the Seebeck coefficient-conductivity measurements is ingenious. The use of spontaneous de-doping process allow to capture the Seebeck coefficient versus conductivity across four orders of magnitude of conductivity, which is good to avoid the sample-to-sample variation in comparison to the commonly used method (i.e. to measure many films doped with different dopant concentrations).

However, the story described here is incomplete, and the data is insufficient to support their point of view. Some needed explanations are also missing. More importantly, there are some scientific inconsistencies, and also clerical errors, making this paper become less readable. At this stage, it is not good to publish this manuscript in Nature Communications. No matter where this manuscript is submitted to, it need to be revised before publication. This study will become more attractive and readable if more supporting data and explanations/clarifications with less inconsistencies are included at which they are desirable. Some of the questions/comments/suggestions are listed as following:

1. Morphology, which is related to most of the discussion and the conclusion, play a key role throughout this paper; therefore, a correct description is very important. However, only KPFM was employed to characterize the microstructure in this study. It is certainly preferable to also characterize the films at least by GIWAXS and etc., to see how they pack before and after doping, as well as what is going on as a function of time during de-doping process. In addition to aggregation and phase separation, packing could be another possibility for the change of transport properties.
2. The Figures in supporting information were completely incorrectly numbered. The number starts from S6, and Figures S1-S5 are missing. As I understand it, actually, Figure S6 should be S1, S7 should be S2, etc. It is strongly recommended to check the whole manuscript more carefully.
3. The transport model was assigned to phonon-assisted hopping in this wrok. Actually, however, there are couple of different transport models which were verified to apply to conducting polymers. There is no any clear explanation for why the transport model is phonon-assisted hopping. Particularly, temperature-dependent electrical conductivity and Seebeck coefficient measurements, which are the most commonly seen data for the analysis of transport model, are absence. This raise a question about the reliability and correctness of the transport model.
4. On page 9, it is claimed that "The thermopower vs. electrical conductivity trend for the annealed sample was similar to the un-annealed polymer that was doped at 25 °C (see Fig. S1), ruling out thermal annealing at 75 °C as the cause of the modified thermopower vs. electrical conductivity trend." It is agreed that the change of charge transport profile cannot be explained by thermal annealing alone, but we do not think the thermal annealing can be completely ruled out because what we can see is a difference in the range of <0.1 and >10 S/cm between the thermopower-conductivity curves for the samples annealed at 75 °C and non-annealed (see "Figure S6"). Of course the thermal annealing is not the main reason, but it could be one of the reasons, for the change in charge transport. It is recommended to revise the sentence on page 9 to be more objective.

5. There might be some scientific inconsistencies about the KPFM measurements. Surface potential contrast (SPC) mapping by KPFM was used to characterize the heterogeneous spatial distributions of dopants and the shape of DoS. The P3HT film doped at 25 °C exhibits an obviously broader DoS than that doped at 75 °C while the SPC profiles of the PDPP4T films are very similar at different doping temperature. This result is very likely to reveal that there are different transport pathways between the two samples of P3HT, and similar pathway between the two samples of PDPP4T. Broad DOS flatten the thermopower-conductivity curve. However, the thermopower-conductivity curves showed a same trend in P3HT films, and an apparently different trend in PDPP4T samples. Such a result is very scientifically inconsistent with the results of KPFM. On the other hand, the KPFM result of the neat PDPP4T should be provided as a reference. Lastly, providing a clear definition and/or physical significance for the width and the value of SPC distribution is very encouraged because they play an important role in the discussion. A clear physical significance is good for readers to understand what this experience talk about.

6. The KPFM measurements described the de-doping processes as a function of time. To associate the KPFM results with the transport properties, additional time-dependent Seebeck coefficient and conductivity measurements of de-doping processes are very preferable to be included.

7. Repeatable experimental results are very important for scientific development. The experimental details of this work are incomplete in both main text and supporting information. For example, film thickness, device architecture (the films were patterned or not), measurement error for thermopower, the measurement method used for electrical conductivity (2-probe, 4-probe, Van der Pauw, or other measurements) and etc. are missing. A detailed experimental section at least should be included in supporting information.

8. To verify the unusual dependence of doping temperature on the thermoelectric properties of PDPP4T, the home-made PDPP4T was characterized. Another good way to verify the unusual dependence might be to provide a sample-to-sample variation.

9. On page 10, the discussion is based on "Figure S4". Actually, however, "Figure S4" is non-existent. As we understand, what they meant could be Figures S9 and/or S10. It is very likely that the readers don't understand what the authors are talking about at all.

10. In Figure 5a, the meaning of "E (eV)" is unclear. Does E represent the difference between Fermi level and transport level ($E_f - E_t$) or represent the energy vs. the vacuum energy level? A clear definition can help the readers to better understand.

Reviewer #2 (Remarks to the Author):

The authors have used two different organic semiconductors (P3HT and PDPP4T) to shed light on the influence of dopant special distribution on the thermoelectric behaviors of organic (polymeric) materials. It is shown that uniformity of dopant distribution in the polymer films is linked to differences in DOS shape, which is further linked to thermoelectric performance (electrical conductivity and Seebeck coefficient). Kelvin Probe Force Microscopy (KPFM) and photoluminescence were used to reveal dopant distribution in iodine doped films. This is a very insightful study and the text is very well written. Good justification is provided for key assertions and great care appears to have been taken to mitigate sources of error in various measurements. This work is very impactful in regard to processing organic thermoelectric materials and likely will have great impact on the broader field of polymer-based electronics. This is excellent work. There are just two relatively minor points for the authors to consider prior to publication:

1. The symbol for the transport parameter "s" could potentially be confused with Seebeck coefficient, which is often associated with "S" (capitalized letter).

2. What is the spatial resolution of the KPFM technique? It does not appear to be explicitly mentioned. Providing a value and a supporting reference would be helpful.

Tuning charge transport dynamics via clustering of dopants in organic semiconductor thin films

Response to Comments of the Reviewers of Ms. No. NCOMMS-18-10570000-T

Reviewers' comments:

Reviewer #1 (Remarks to the Author):

This study described a story about the effects of aggregation of dopant on charge transport in conducting polymers. The morphology of doped films could be tuned by doping temperature. The aggregation of dopant broadened the distribution of DoS, and hence affecting charge transport. The spatial distribution of dopant was highlighted to be an important role in thermoelectrics and other electronic applications. The novelty of this work is to probe the spatial heterogeneity and the energy distribution by using KPFM, which provides an idea on characterization for future study. Additionally, the design of the Seebeck coefficient-conductivity measurements is ingenious. The use of spontaneous de-doping process allow to capture the Seebeck coefficient versus conductivity across four orders of magnitude of conductivity, which is good to avoid the sample-to-sample variation in comparison to the commonly used method (i.e. to measure many films doped with different dopant concentrations). However, the story described here is incomplete, and the data is insufficient to support their point of view. Some needed explanations are also missing. More importantly, there are some scientific inconsistencies, and also clerical errors, making this paper become less readable. At this stage, it is not good to publish this manuscript in Nature Communications. No matter where this manuscript is submitted to, it need to be revised before publication. This study will become more attractive and readable if more supporting data and explanations/clarifications with less inconsistencies are included at which they are desirable. Some of the questions/comments/suggestions are listed as following:

1. Morphology, which is related to most of the discussion and the conclusion, play a key role throughout this paper; therefore, a correct description is very important. However, only KPFM was employed to characterize the microstructure in this study. It is certainly preferable to also characterize the films at least by GIWAXS and etc., to see how they pack before and after doping, as well as what is going on as a function of time during de-doping process. In addition to aggregation and phase separation, packing could be another possibility for the change of transport properties.

Response: We agree with the reviewer that morphology plays an important role in terms of how dopants are distributed in the films. We have obtained wide angle x-ray scattering (WAXS) data for P3HT and PDPP4T and they are now included as Figure S8 and Figure S9 in the supporting information. We have added the following paragraphs to the manuscript.

The WAXS data for pristine, unannealed, pristine P3HT showed two signature peaks at q values of 0.37 ($d_{100}=16.98 \text{ \AA}$), and 1.65 \AA^{-1} ($d_{020}=3.81 \text{ \AA}$) (see Figure S8). Upon annealing at $75 \text{ }^\circ\text{C}$, the peaks shift slightly and appear at $q=0.38$ ($d_{100}=16.53 \text{ \AA}$) and 1.67 \AA^{-1} ($d_{020}=3.71 \text{ \AA}$). For unannealed, pristine PDPP4T, we observed two signature peaks at q values of 0.30 ($d_{100}=20.94 \text{ \AA}$), and 1.66 \AA^{-1} ($d_{020}=3.79 \text{ \AA}$). For films annealed at $75 \text{ }^\circ\text{C}$, the peaks appear at q values of 0.30 ($d_{100}=20.94 \text{ \AA}$), and 1.67 \AA^{-1} ($d_{020}=3.76 \text{ \AA}$). Both polymers had broad peak around $q=1.25 \text{ \AA}^{-1}$ attributed to the amorphous phase. No additional peaks appeared upon annealing.

The WAXS pattern of the P3HT film doped at $25 \text{ }^\circ\text{C}$ was identical to the pattern of pristine P3HT films (See Figure S9a). The WAXS pattern of the de-doped film was also identical to the pattern of pristine films. The WAXS pattern of P3HT films doped at $75 \text{ }^\circ\text{C}$ shows peaks at $q=0.35$ ($d_{100}=17.97 \text{ \AA}$) and at 1.73 \AA^{-1} ($d_{020}=3.63 \text{ \AA}$) indicating that the dopants may have penetrated the crystalline domains (Figure S9b). The broad peak around $q=1.25 \text{ \AA}^{-1}$ also narrowed and has a pronounced feature. The WAXS patterns of PDPP4T films doped at room temperature and dedoped films were identical to pristine PDPP4T (Figure S9c). The WAXS pattern of P3HT films doped at $75 \text{ }^\circ\text{C}$ shows peaks at $q=0.28$ ($d_{100}=22.44 \text{ \AA}$) and at 1.68 \AA^{-1} ($d_{020}=3.74 \text{ \AA}$) (Figure S9c). The shifts were smaller in the WAXS patterns of PDPP4T films compared to P3HT films doped at different temperatures.

We feel that mapping the location of the dopants within the polymer films is important but a thorough structural investigation is beyond the scope of this manuscript and will be the subject of our investigation in the near future.

2. The Figures in supporting information were completely incorrectly numbered. The number starts from S6, and Figures S1-S5 are missing. As I understand it, actually, Figure S6 should be S1, S7 should be S2, etc. It is strongly recommended to check the whole manuscript more carefully.

Response: We apologize for this error and we have fixed it. Since the word file attached with the manuscript does have all the files, we suspect that this error occurred when the word file was converted to PDF. We have directly uploaded a PDF file to avoid this problem.

3. The transport model was assigned to phonon-assisted hopping in this work. Actually, however, there are couple of different transport models which were verified to apply to conducting polymers. There is no any clear explanation for why the transport model is phonon-assisted hopping. Particularly, temperature-dependent electrical conductivity and Seebeck coefficient measurements, which are the most commonly seen data for the analysis of transport model, are absence. This raise a question about the reliability and correctness of the transport model.

Response: We thank the referee for pointing this out; we do, in fact, contrast our data and the hopping model to the band model of Snyder and Kang (Fig. 5d, S4 and

S6). We consistently find the hopping model, which has been widely used to understand charge transport in polymers (Ref. 52-56 in the manuscript, as well as Tessler, N. et al., *Advanced Materials* 21, 2741–2761 (2009), to cite a few). This is particularly true for polymers that have structural disorder or are amorphous, including P3HT and PEDOT (Ihnatsenka, S. et al., *Phys. Rev. B* 92, 035201 (2015), Abdalla, H. et al., *Phys. Rev. B* 96, 241202(R) (2017), Vukmirović, N. & Wang, L. W., *Nano Letters* 9, 3996–4000 (2009), Lu, N. et al., *Organic Electronics* 29, 27-32 (2016), Lu, N. et al., *Phys. Chem. Chem. Phys.* 18, 19503–19525 (2016)) where phonon-assisted hopping was found to agree more thoroughly with the experimental data. We delve deeper into the comparison of the various parameters of the hopping model, including Miller-Abrahams vs. Marcus (polaronic binding) hopping rates, in a theory paper that is currently under review (Ref. 56). A preprint is available on arXiv: <https://arxiv.org/abs/1901.03370>

We agree that temperature-dependent measurements could provide further points of comparison between the model and experiments. Unfortunately, we do not have temperature-dependent data available at this time because of the de-doing kinetics. We are in the process of controlling this kinetics and the results of this investigation will be reported in due course. Nonetheless, the phonon-assisted hopping model has already been compared to temperature-dependent measurements on various polymers including P3HT and PEDOT (Lu, N. et al., *Phys. Chem. Chem. Phys.* 18, 19503–19525 (2016), Mendels, D. & Tessler, N., *J. Phys. Chem. Lett.* 5, 3247–3253 (2014), Ihnatsenka, S. et al., *Phys. Rev. B* 92, 035201 (2015), Abdalla, H. et al., *Scientific Reports* 5, 16870 (2015), Abdalla, H. et al., *Phys. Rev. B* 96, 241202(R) (2017)) with very close agreement across a broad range of temperatures.

4. On page 9, it is claimed that “The thermopower vs. electrical conductivity trend for the annealed sample was similar to the un-annealed polymer that was doped at 25 °C (see Fig. S1), ruling out thermal annealing at 75 °C as the cause of the modified thermopower vs. electrical conductivity trend.” It is agreed that the change of charge transport profile cannot be explained by thermal annealing alone, but we do not think the thermal annealing can be completely ruled out because what we can see is a difference in the range of <0.1 and >10 S/cm between the thermopower-conductivity curves for the samples annealed at 75 °C and non-annealed (see “Figure S6”). Of course the thermal annealing is not the main reason, but it could be one of the reasons, for the change in charge transport. It is recommended to revise the sentence on page 9 to be more objective.

Response: We thank the reviewer for pointing this out and we have revised the line and it now reads as “indicating that thermal annealing at 75 °C alone may not be cause of the modified α vs. σ trend.”

5. There might be some scientific inconsistencies about the KPFM measurements.

Surface potential contrast (SPC) mapping by KPFM was used to characterize the heterogeneous spatial distributions of dopants and the shape of DoS. The P3HT film doped at 25 °C exhibits an obviously broader DoS than that doped at 75 °C while the SPC profiles of the PDPP4T films are very similar at different doping temperature. This result is very likely to reveal that there are different transport pathways between the two samples of P3HT, and similar pathway between the two samples of PDPP4T. Broad DOS flatten the thermopower-conductivity curve. However, the thermopower-conductivity curves showed a same trend in P3HT films, and an apparently different trend in PDPP4T samples. Such a result is very scientifically inconsistent with the results of KPFM. On the other hand, the KPFM result of the neat PDPP4T should be provided as a reference. Lastly, providing a clear definition and/or physical significance for the width and the value of SPC distribution is very encouraged because they play an important role in the discussion. A clear physical significance is good for readers to understand what this experience talk about.

Response: We thank the reviewer for raising this issue. We have now added the SPC of neat PDPP4T to Figure 3. In KPFM, we measure the energy location of the Fermi level not the width of the DoS, which we show in Figure 3. In principle, we can couple this formation with other parameters such as carrier concentration to map the width of DoS. In analyzing the α vs. σ curve, we have to take in to account the width of the DoS *and* the shape of the DoS. As the reviewer points out, changing the width of the DoS, while maintaining a Gaussian DoS, will simply shift the α vs. σ on the log-log curve with minimal difference in its shape. The shape of the α vs. σ curve does depend on the shape of DoS (Gaussian vs. heavily tailed Gaussian). Our hypothesis is that the shape of the DoS depends on the distribution of dopants in the film. Clustering of dopants leads to a heavily-tailed Gaussian whereas homogeneous distribution leads to a Gaussian DoS. We used SPC profiles to probe the clustering of the dopants. In P3HT, irrespective of the doping temperature (25 °C or 75 °C), we find that SPC profiles are broad. The SPC profiles also indicate that the films doped at 75 °C have less clustering compared to films doped at 25 °C. In both cases, fitting the experimental α vs. σ curve does require the use of a heavily-tailed Gaussian DoS. When compared with P3HT, SPC profiles of doped PDPP4T films are narrow. However, the fitting the experimental α vs. σ curve for films doped at 25 °C does require use of a heavily-tailed Gaussian indicating that there is some clustering of dopants in these samples.

6. The KPFM measurements described the de-doping processes as a function of time. To associate the KPFM results with the transport properties, additional time-dependent Seebeck coefficient and conductivity measurements of de-doping processes are very preferable to be included.

Response: We thank the reviewer for this comment. All of the experimental α vs. σ curves included in the manuscript were recorded during the de-doping process over time.

7. Repeatable experimental results are very important for scientific development. The experimental details of this work are incomplete in both main text and supporting information. For example, film thickness, device architecture (the films were patterned or not), measurement error for thermopower, the measurement method used for electrical conductivity (2-probe, 4-probe, Van der Pauw, or other measurements) and etc. are missing. A detailed experimental section at least should be included in supporting information.

Response: We thank the reviewer for pointing this out, and have now updated the Methods section in the main text of the paper. We have included the device architecture details, film thickness, and the four-probe measurement.

8. To verify the unusual dependence of doping temperature on the thermoelectric properties of PDPP4T, the home-made PDPP4T was characterized. Another good way to verify the unusual dependence might be to provide a sample-to-sample variation.

Response: As stated in the manuscript, we have done the experiments with both commercial PDPP4T and home-made PDPP4T. We observed similar behavior in both samples. The α vs. σ data from commercial PDPP4T is in the main manuscripts whereas the α vs. σ for homemade PDPP4T is in the supporting information (Fig S3). The manuscript has the following line to alert the reader “To verify the unusual dependence of doping temperature on the thermoelectric properties of PDPP4T, we repeated the doping and measurement at temperatures of 25 °C and 75 °C using PDPP4T synthesized in-house, and observed similar behavior (see Fig. S3).”

9. On page 10, the discussion is based on “Figure S4”. Actually, however, “Figure S4” is non-existent. As we understand, what they meant could be Figures S9 and/or S10. It is very likely that the readers don’t understand what the authors are talking about at all.

Response: We apologize for this error and we have fixed it. Since the word file attached with the manuscript does have all the files, we suspect that this error occurred when the word file was converted to PDF. We have directly uploaded a PDF file to avoid this problem.

10. In Figure 5a, the meaning of “E (eV)” is unclear. Does E represent the difference between Fermi level and transport level ($E_f - E_t$) or represent the energy vs. the vacuum energy level? A clear definition can help the readers to better understand.

Response: The meaning of E is the carrier energy in electron volts. In all cases, we chose the zero of energy $E=0$ to be in the middle of Gaussian distribution for the density of states before doping g_i given by $g(i) = \frac{1}{2\pi\Gamma_E^2} \exp\left(-\frac{E^2}{2\Gamma_E^2}\right)$, which has now been added in **Materials and Methods**. We use this as a reference point throughout for consistency, as depicted in Figure 5a.

In all cases, we first find the position of the Fermi level E_f relative to this reference energy. Then we solve the Pauli master equation to find the carrier distribution, from which we compute the transport energy E_t ; lastly, the Seebeck coefficient is computed from the difference $E_f - E_t$. Because of the Seebeck coefficient depending on the difference between Fermi and transport energy, the energy “zero” reference point cancels and does not affect the results. Another useful feature of this choice is that when $E_f=0$ (the Fermi level is in the middle of the Gaussian), then the states above and below E_f are symmetric, which leads to $E_t=E_f=0$ and the Seebeck goes to zero due to cancellation between electron (above E_f) and hole (below E_f) states. One of the key findings of our work is that the DOS is modified by doping, which makes it asymmetric and increases the magnitude of the Seebeck coefficient.

Reviewer #2 (Remarks to the Author):

The authors have used two different organic semiconductors (P3HT and PDPP4T) to shed light on the influence of dopant special distribution on the thermoelectric behaviors of organic (polymeric) materials. It is shown that uniformity of dopant distribution in the polymer films is linked to differences in DOS shape, which is further linked to thermoelectric performance (electrical conductivity and Seebeck coefficient). Kelvin Probe Force Microscopy (KPFM) and photoluminescence were used to reveal dopant distribution in iodine doped films. This is a very insightful study and the text is very well written. Good justification is provided for key assertions and great care appears to have been taken to mitigate sources of error in various measurements. This work is very impactful in regard to processing organic thermoelectric materials and likely will have great impact on the broader field of polymer-based electronics. This is excellent work. There are just two relatively minor points for the authors to consider prior to publication:

Response: We thank the reviewer for the positive comments. We address the reviewers concerns below.

1. The symbol for the transport parameter "s" could potentially be confused with Seebeck coefficient, which is often associated with "S" (capitalized letter).

Response: We concur with the reviewer that the transport parameter's' can be confused with Seebeck coefficient 'S'. However, this is not our choice. We are using the symbol used by Snyder and co-workers for their model. To avoid confusion, we have used α as the symbol for Seebeck co-efficient in the manuscript.

2. What is the spatial resolution of the KPFM technique? It does not appear to be explicitly mentioned. Providing a value and a supporting reference would be helpful.

Response: We chose the lift height (40 nm) and scan dimension (2 μm x 2 μm) to maximize spatial resolution. Under these conditions, spatial resolution is limited by tip radius, which is ~ 40 nm.

Reviewer #1 (Remarks to the Author):

The authors have been resolved most of the issues pointed out last time and revised the main text to improve the readability of the manuscript. Now this could be an insightful study for understanding the structure-property relationship of conducting polymers. There are two minor comments for consideration prior to publication.

1. Both the P3HT and PDPP4T systems doped at 75 °C showed an increase in lamellar distance while the p-p distance decreased in the doped P3HT system and was fairly the same in the doped PDPP4T film in comparison to the pristine polymer films. This result indicates that the dopants/counterions might mainly reside in the region of alkyl side chain, and such a phenomenon has been reported in some recent reports (see below). It is encouraged to mention this point with relevant references.

a) 2D coherent charge transport in highly ordered conducting polymers doped by solid state diffusion, DOI: 10.1038/NMAT4634

b) X-Ray Scattering Reveals Ion-Induced Microstructural Changes During Electrochemical Gating of Poly(3-Hexylthiophene), DOI: 10.1002/adfm.201803687

2. Page 11, there is a sentence "The WAXS pattern of P3HT films doped at 75 °C shows peaks at $q = 0.28$ ($d_{100} = 22.44 \text{ \AA}$)", where "P3HT" could be "PDPP4T" according to the details shown in Figure S9d.

Reviewer #2 (Remarks to the Author):

The authors did a good job of addressing reviewer comments. The manuscript should be accepted.

Tuning charge transport dynamics via clustering of dopants in organic semiconductor thin films

Response to Comments of the Reviewers of Ms. No. NCOMMS-18-10570000A

Reviewer #1:

1. Both the P3HT and PDPP4T systems doped at 75 °C showed an increase in lamellar distance while the p-p distance decreased in the doped P3HT system and was fairly the same in the doped PDPP4T film in comparison to the pristine polymer films. This result indicates that the dopants/counterions might mainly reside in the region of alkyl side chain, and such a phenomenon has been reported in some recent reports (see below). It is encouraged to mention this point with relevant references.

a) 2D coherent charge transport in highly ordered conducting polymers doped by solid state diffusion, DOI: 10.1038/NMAT4634

b) X-Ray Scattering Reveals Ion-Induced Microstructural Changes During Electrochemical Gating of Poly(3-Hexylthiophene), DOI: 10.1002/adfm.201803687

Our response: We thank the reviewer for these references and we have added them as Refs 50 and 51.

2. Page 11, there is a sentence “The WAXS pattern of P3HT films doped at 75 °C shows peaks at $q=0.28$ ($d_{100}=22.44 \text{ \AA}$)”, where “P3HT” could be “PDPP4T” according to the details shown in Figure S9d.

Our response: We thank the reviewer for pointing this error. We replaced ‘P3HT’ with ‘PDPP4T’

Reviewer 2 did not request any changes. We thank both the reviewers for the constructive criticism and feedback. Their reviews vastly improved the quality of our manuscript